# Breaking down hierarchies of decision-making in primates

**Alexandre Hyafil[1]\*, Rubén Moreno-Bote[1,2]**

[1]CBC, DTIC, Universitat Pompeu Fabra, Barcelona, Spain; [2]Serra Húnter Fellow Programme, Universitat Pompeu Fabra, Barcelona, Spain

**Abstract** Possible options in a decision often organize as a hierarchy of subdecisions. A recent study concluded that perceptual processes in primates mimic this hierarchical structure and perform subdecisions in parallel. We argue that a flat model that directly selects between final choices accounts more parsimoniously for the reported behavioral and neural data. Critically, a flat model is characterized by decision signals integrating evidence at different hierarchical levels, in agreement with neural recordings showing this integration in localized neural populations. Our results point to the role of experience for building integrated perceptual categories where sensory evidence is merged prior to decision.

\*For correspondence: alexandre. hyafil@gmail.com

**Competing interests:** The authors declare that no competing interests exist.

## Introduction

Coffee with Jules (if so, which cafe?) or cinema with Jim (if so, what movie?) ? A recent study by Lorteije and colleagues investigated how perceptual mechanisms implement such hierarchically-structured decisions that fill up our daily lives (*Lorteije et al., 2015*). Monkeys performed saccades to one of 4 possible targets based on the information provided at the primary branching and at secondary branching points (the 'correct' and 'incorrect' branching points) leading to the targets (*Figure 1A*). Patterns of responses convincingly indicated that monkeys can integrate information from all branching points in parallel. But does the decision space mimic the hierarchical organization, with parallel decisions going on (about Jules/Jim, coffee place, movie) (*Figure 1B*), or does it directly compare final options (with Jules at Moe's vs. Sicario with Jim vs. etc.)(*Figure 1C*)? Lorteije and colleagues report two behavioral and two neural effects that they argue speak unanimously in favor of the former hierarchical model against the latter non-hierarchical 'flat model' of decision-making. We show in contrast that all four effects can equally (and more parsimoniously) be explained by the flat model of decision-making, which is also perfectly compatible with the new effects from the same dataset described in the companion paper by Zylberberg and colleagues (*Zylberberg et al., 2017*).

## Results

First, stimulus difficulty at the primary branch (L1) was shown to have no influence on the performance at the secondary branch (L2), a result that was replicated by a race-model implementation of hierarchical decision-making, but not of the flat decision-making. However, this result seems a particular feature of their specific choice of implementation of the race model (*Vickers, 1979*; *Drugowitsch et al., 2014*), which included neither inhibition between option representations, nor activation rectification (i.e. enforcing non-negative unit activation), two classical ingredients of race models (*Usher and McClelland, 2001*; *Tsetsos et al., 2012*; *Churchland and Ditterich, 2012*). Inhibition between option representations and rectification may underlie the well-known reduction of choice-related neural activity when more choice alternatives are provided (*Churchland and Ditterich, 2012*). We simulated a race model implementation of the flat model with both rectified

**eLife digest** Should you go for coffee with Jules, or go to the movie theater with Jim? Both options require you to make additional decisions, for example, which café would you go to, or what movie could you see? Many of our day-to-day decisions have multiple layers of sub-decisions embedded within them that are not necessarily independent. Our opinions of the cafés in town and the movies showing at the theater may influence our decision over whom to spend the afternoon with.

In 2015, researchers at the Netherlands Institute for Neuroscience performed experiments in macaques to try to work out how the brain makes these decisions. The monkeys learned to choose between two visual stimuli (decision 1). The outcome of decision 1 determined whether the animals then had to make decision 2 or decision 3. The results suggested that the monkeys initially made all three comparisons independently and in parallel, before combining the evidence to select their overall strategy. This process is referred to as hierarchical decision-making. In the analogy above, one would compare the relative merits of Jules versus Jim, café A versus café B, and a horror movie versus a comedy at the same time before deciding what to do.

Hyafil and Moreno Bote have now reanalyzed the data published in 2015 using new computer simulations. This second analysis suggests the results are in fact more consistent with an alternative model of decision-making called a flat model, in which the brain compares all of the final options simultaneously (Jules + café A; Jules + café B; Jim + horror movie; Jim + comedy) before choosing between them.

Making decisions by comparing the final outcomes becomes easier as the brain learns through experience to associate stimuli that often occur together. Hyafil and Moreno Bote hypothesize that in response to a new situation, the brain may sometimes start off by using hierarchical decision-making before switching to a more accurate flat model as experience allows.

In response to the findings of Hyafil and Moreno Bote, the researchers who conducted the work reported in 2015 have also reanalyzed the original data, and carried out a new experiment in human volunteers. They argue that the flat model provides a poor fit to the original data and struggles to explain the new data. Future studies can build on these conflicting findings by further exploring the limits of parallel decision-making, which may help us to understand how the brain is able to make multiple decisions while keeping the future consequences in mind.

---

activation, cross-inhibition and self-excitation, whereby each of the four options competed during accumulation of evidence (*Figure 1D*, see Material and methods). Self-excitation models the instability present at the initial point of the race in attractor models of decision-making (*Roxin and Ledberg, 2008*) and can be at the heart of well-known urgency signals to speed up decisions (*Drugowitsch et al., 2012*). Noise in the model did not scale with stimulus intensity, in line with recent results suggesting that noise in perceptual accumulation tasks is associated with the accumulation rather than with the sensory process (*Drugowitsch et al., 2016*), and as it is typically assumed in drift-diffusion models of decision making (*Gold and Shadlen, 2007*). We used a non-absorbing decision threshold (i.e. activity after the decision as not bounded), but found the same results for simulations with an absorbing bound. Parameters were tuned to reproduce qualitatively the proportion for each of the four possible response types, as well as the impact of each sample in the stream and the psychometric curve for both L1 and L2 decision (*Figure 2—figure supplement 2A–D*). The impact of L1 difficulty onto L2 for the flat model indeed disappeared when we used this common implementation of the race model (*Figure 2—figure supplement 2E*).

In the companion paper (*Zylberberg et al., 2017*), Zylberberg and colleagues question the generality of this finding, arguing that it may not be compatible with the pattern of short reaction times they provide in a new analysis for the same monkey dataset. However, our mathematical analysis (Appendix 1) shows that a clear influence of L1 difficulty onto L2 performance emerges principally when L1 difficulty strongly modulates reaction times, and thus the time of integration of L2 evidence. Short reaction times with low modulation by task difficulty as described in the companion article are thus perfectly compatible with lack of influence of L1 difficulty onto L2 performance. This result was

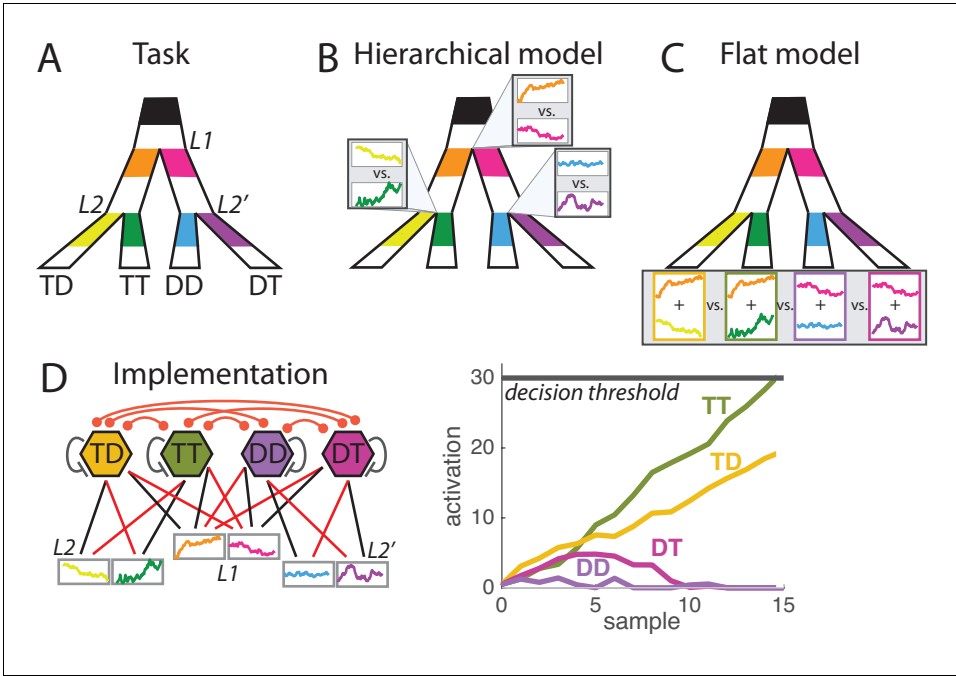

**Figure 1.** Hierarchical vs. flat models of perceptual decision during a hierarchically-structured visual task. (**A**) Structure of the task. At each trial monkeys must detect the correct option out of four possible responses based on the visual information provided at the primary branching point L1 and at two secondary branching points L2 and L2'. Visual information consists of segments of flickering luminosity at the start of each branch (color segments in our depiction; visual samples changing every 50 ms for a total period of 1000 ms). Animals must make a saccade towards the final point that passes through branches of maximal luminosity; that is, must decide based on information scattered across the visual field. Each of the four responses is categorized as TT, TD, DT or DD depending on whether it corresponds to a target T (correct branch) or distractor D (incorrect branch) at first and second branching points. Detailed description of the task can be found in [*Lorteije et al., 2015*]. (**B**) Hierarchical decision model of perception. In the hierarchical model, parallel decision processes run at each branching point (L1, L2 and L2') and are integrated into a motor response at a later stage. It can be implemented as a race model composed of three races, each with two possible sub-choices. (**C**) Flat decision model of perception. In the flat model, the decision space is composed of the four possible final responses, so for each response the animal must sum the information provided at the corresponding primary and secondary branches (here depicted by the sum of the two luminosity signals). (**D**) Implementation of the flat decision model. (Left panel) Four units coding for the four possible responses integrate information from both L1 and L2 branches leading to that response, as represented by the pattern of connections from sensory units (coding for instantaneous sensory value). Connectivity include self-excitation as well as homogeneous cross-inhibition between all units. Black and red arrows indicate respectively excitatory and inhibitory connections. (Right panel) Simulation for one trial, depicting the activity of each unit across time during perceptual integration. Activity is bounded to positive values (rectification). When activity of one unit reaches the decision threshold, the related response is selected.

indeed reproduced in a simulation where the threshold was lowered to produce shorter reaction times, and the boundary was eliminated for the first 10 samples (*Figure 2A–F*). This choice of the threshold implements a form of time-dependent, collapsing bound (*Churchland and Ditterich, 2012*; *Drugowitsch et al., 2012*) and it is consistent with the minimum viewing time imposed on monkeys (*Figure 2*). Without such time-dependent bound, the model can produce either a large proportion of premature responses or very large mean reaction times. The discrepancy between results from these simulations with those performed in the companion paper (who do find dependency of L2 performance on L1 difficulty with short reaction times) may emerge from early responses (<500 ms) in easy trials that are present in their simulations, but prohibited in our simulations by the absence of decision boundary for the first 10 samples.

The second interesting observation from monkey behavior reported by Lorteije and colleagues was that L1 decisions were biased towards the branch that leads to the easier L2 decision. The effect

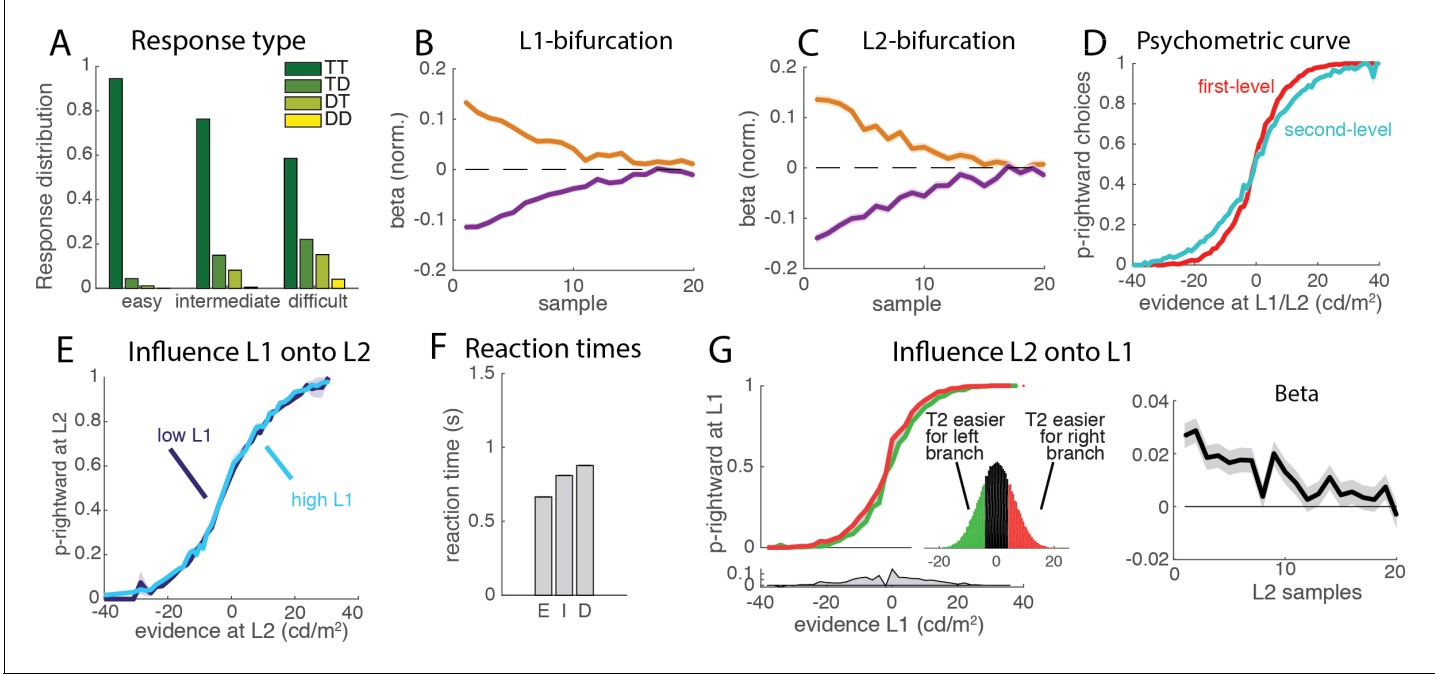

**Figure 2.** Behavioral properties of the flat race model reproduce original monkey data. The figures presented here correspond to a parameter set adjusted to reproduce reaction times (see Introduction). (**A**) Response type for the flat model. We simulated a simple race model with four possible options that compete for selection, based on a standard implementation of race models [***Drugowitsch et al., 2014***]. The histogram shows the distribution of each of the four response types (TT, TD, DT, DD) for each level of trial difficulty from simulations of the race implementation of the flat model. (**B**) Influence of luminance fluctuations on the L1 decisions. Weights for each of the target (orange) and distractor (blue) samples across a trial for L1 decisions, as measured with logistic regression. The model reproduces the primacy effect observed in monkey behavioural data, whereby earlier samples have larger influence on L1 choices than later samples. Shades indicating 95% confidence intervals (barely visible) are nearly collapsed to the main line. Weights have been normalized. (**C**) Influence of luminance fluctuations on the L2 decisions. Legend same as B. (**D**) Psychometric curve. The curve represents the probability that the right segment was selected at the first (L1, red curve) and second (L2, light blue curve) level depending on the strength of the evidence in favor or right vs. left path at the corresponding branching point. Steeper curve indicates better performance at the primary than secondary branches. (**E**) Influence of L1 difficulty onto L2 performance. Psychometric curve for L2 decision was computed separately for easy and difficult L1 trials. Unlike the flat model implemented in the original study, we find no difference in L2 performance (permutation test; p>0.4), in accordance with observed animal behavior. The null effect emerges because the minimum accumulation time before reaching a decision impedes early response even in the presence of strong evidence in L1 (see Supp. Material). When inhibition was removed, a significance interaction was recovered (***Figure 2—figure supplement 1A***). (**F**) Mean reaction times. Reaction times for the 3 types of difficulty level (E: easy. I: intermediate, D:difficult) reproduce those reported in the companion paper by Zylberberg and colleagues (***Zylberberg et al., 2017***). (**G**) Influence of L2 and L2' difficulty onto L1 choice. Psychometric curve is plotted separately for trials where decision is easier at the left than right secondary branch (green curve), and where decision is easier at right than left secondary branch (red curve). Inset represents the distribution of difference in evidence between the two branches. The effect emerges because strong evidence at one secondary branch will bias the race towards selecting the corresponding final option, thus appearing as a bias in L1 decision towards selecting the primary branch leading to this secondary branch. Information provided at secondary branches biases choice at L1 towards selecting the branch that leads to easier secondary branch, as observed in animal behavior. Lower panel represents the difference between the two psychometric curves. (Right Panel): Influence of difficulty of L2 on the L1 decision across time, as estimated from the logistic regression. Grey shades indicate 95% confidence intervals. L1 choice is affected by difference in perceptual difficulty between L2 and L2' branches in early visual samples.

The following figure supplements are available for figure 2:

**Figure supplement 1.** Reciprocal influence of L1 and L2 evidence and decisions.

**Figure supplement 2.** Behavioral and neural effects for a parameter set yielding longer reaction times than in monkey data.

could only be explained in the hierarchical model by invoking an extra modulatory signal passed on from L2 to L1 that must be carefully tuned (and was not implemented in the race model of [***Lorteije et al., 2015***]). By contrast, such effect is readily accounted for by the flat model, as a strong signal at a secondary branch will boost the chances of selecting the corresponding final option, and

thus bias towards selecting the L1 path leading to this option. Indeed, the effect was reproduced in the simulations (*Figure 2G*). This relates to one important advantage of the flat model over the hierarchical one: the flat model, by integrating the strength of evidence from each level, takes optimally into account uncertainty in each of the decisions, leading to more accurate responses. By contrast, the hierarchical model integrates decisions at each level independently of the level of evidence in support of each decision. This can only be amended by *ad hoc* mechanisms such as that proposed by Lorteije and colleagues, that probably do not scale up adequately when more than two options are available at each level.

At the physiological level, neural groups in visual cortex integrated information conveyed at the primary and secondary branching points leading to the path in their receptive field. Selection signals were shown to first differentiate options at the level of secondary branches irrespective of whether the branch is the correct or incorrect one (L2 or L2′ branching), and subsequently grow larger for the L2 than for L2′ branching. In the hierarchical model, this can only be explained by referring to a modulating signal once decision is reached in L1 that differentially modulates the L2 race signals corresponding to the selected and non-selected L1 branches. In the flat model, such dynamics emerges naturally with activation rectification, because activation corresponding to the two incorrect options in the incorrect L1 branch vanish as L1 signals favor the two alternative options, and so their difference is also reduced (*Figure 3A*). Indeed, activation rectification plays a role in the dynamics of recorded neural responses, that reach a floor value (0) in the final part of the integration period in non-selected branches (Figure 4 of [*Lorteije et al., 2015*]). Finally, an interaction between L2 and L2′ selection signals was reported: clear evidence in favor of one of the two options in L2 reduced the selection signal in L2′, and vice versa. In the hierarchical model, this would require complex modulatory signals passing on from L2 to L1 to L2′. By contrast, both effects are observed in the flat model (*Figure 3B*): strong evidence for one option will decrease the activation of all other three options through inhibition, increase their chance of collapsing at the zero boundary, and thus reduce the selection signal in the opposite secondary branch.

Overall we show that all four observations that were taken as evidence in favor of the hierarchical model could be accounted for with a standard race-model implementation of the flat model. These observations were robust and did not rely on fine tuning of the parameters. *Table 1* summarizes how each of these observed properties depend on the features of the model. Note that all these features are classical constituents of race models and not ad hoc mechanisms. Overall the flat model provides a more parsimonious explanation of the data, as it does not require appending modulatory signals between parallel decisions as the hierarchical model does.

These behavioral and neural effects reported in the original study provide however no univocal evidence in favor of either model. One fundamental difference between the two is indeed that selection signals in the flat model mix evidence from both L1 and L2 branches, while the hierarchical model predicts unmixed selection signals (the influence across branches can only occur posterior to decision). [At this point, an important distinction has to be made in the flat model between localized activations, which indeed mix evidence from both branches, and selection signals at L2, extracted by looking at the difference between two units in the same L1 branch, and which as shown above are largely insensitive to L1 signals]. In the companion paper, Zylberberg et al. now provide data showing that selection signals in at least 3 out of 4 L2 branches mixes evidence from both L1 and L2 branches (their Figure 4). We believe this observation is most compatible with the flat model and by itself rules out the hierarchical model that relies on complete neural segregation of integration of L1 and L2 evidence (although the possibility remains that these level-mixing integrative neural signals are completely non-causal in monkey decisions). Our simulations reproduce these effects: signals in TT is larger for stronger L1 evidence, while signals in DT and DD are weaker ($p < 10^{-9}$, *Figure 3C*; see Also Appendix 2). As pointed out by Zylberberg and colleagues, our simulations also display a positive modulation in TD signals, unlike the null effect found in monkey data. However, despite a single source of inhibition, the modulation was not equally strong in all four branches: indeed, it was by far the weakest precisely in the TD branch (*Figure 3C*). This weaker effect may have explained the lack of significance in monkey data. Moreover, modulation in TD could be reduced or abolished if the flat model implied stronger inhibition between options related to the same L1 branch (TT-TD and DT-DD) than between options related to distinct L1 branches (e.g. TT-DT, TT-DD). Stronger inhibition between local circuits is indeed a general pattern of cortical connectivity (*Douglas and Martin, 2004*; *Lund et al., 2003*).

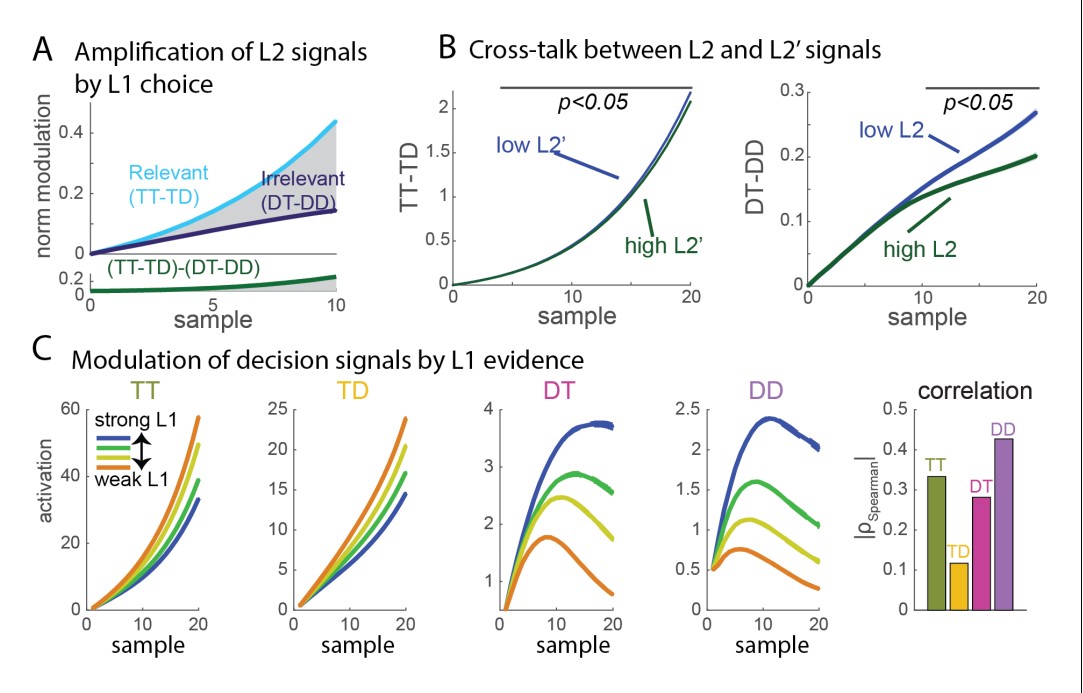

**Figure 3.** Signal properties of the flat race model reproduce original monkey neural data. (**A**) Amplification of L2 signals by L1 choice. Decision signals for L2 (resp. L2'), depicted here in cyan (resp. dark blue) curves, correspond to the difference between the activation of the units coding for the target and distractor in that branch, i.e. TT-TD (resp. DT-DD). Lower panel represents the difference between the two selection signals, i.e. (TT-TD)-(DT-DD). While the two decision signals first increase in parallel, the selection signal rapidly turns much larger for the correct than for the incorrect branches, due to the inhibition between activation of options across different branches. Dynamics closely resembles that observed in neural multi-unit activity in V4. S. e.m. are too small to be visualized. (**B**) Cross-talk between L2 and L2' selection signals. Selection signal for L2 was reduced when the first visual samples in L2' provide clear information in favour of one option (left panel, blue curve for lowerL2' evidence, green curve for higher L2' evidence). The effect readily emerges in the flat model because of inhibition between options across the different branches. The converse effect of L2 difficulty impacting L2' selection signal was also observed (right panel), and both effects reproduce observation from monkey visual cortex. S.e.m. are too small to be visualized. The horizontal bar indicates samples with significant activity difference (t-test, p<0.05). (**C**) Selection signals integrate L1 level of evidence. Unit activity is modulated by evidence at both L1 and L2 levels. Mean activity in each four units (classified as TT, TD, DT, DD) for 4 quartiles of strength of L1 evidence. TT and TD units show positive modulations with L1 evidence, while DT and DD units display negative modulations, consistent with the fact that larger L1 evidence biases competition towards the two options in the correct L1 branch. Right panel displays absolute value of Spearman's rho correlation between strength of evidence and mean activity for all four branches.

Zylberberg and colleagues produced a last analysis of the original dataset, showing that in the flat model errors in L1 are associated with higher sensitivity at L2, whereas monkeys display no such effect. While we acknowledge that this result challenges the current implementation of the flat model, it is at this point equally unknown whether the hierarchical model proposed by the authors could avoid this feature (the reward maximization introduced in the hierarchical model to account for the influence of L2 bias onto L1 choice may produce the same interaction).

## Discussion

It is unclear whether selection signals recorded in visual cortex are generated locally or reflect feed-back process from higher regions where integration of evidence takes place. The flat model is argu-ably more consistent with the latter hypothesis, as it implies integration from and inhibition across distant locations in visual field, which are more typically associated with higher cortical regions (*Wimmer et al., 2015*). In any case, these selection signals certainly represent neural markers for a specific integration process along one subbranch.

One important limitation of flat models is they require to learn high-order representations of the environment that can appropriately integrate evidence from all relevant sensory sources, i.e. the structure of connectivity described in *Figure 1D* may take time to be acquired (*Garrard et al.,*

**Table 1.** Each of reported behavioral and neural effects and the associated features from the flat decision model required to display such effects.

| Effect | Required model features |
| --- | --- |
| Independence of L2 from L1 difficulty | Signal-independent noise, low dependence of RTs on L1 difficulty |
| Bias of L1 choice by L2 difficulty | none |
| Amplification of L2 signals by L1 choice | Activation rectification |
| Cross-talk between L2 and L2' signals | Activation rectification |

*1997*; *McClelland and Rumelhart, 1985*). Here, monkeys performed ~60,000 trials each, possibly long enough to learn appropriate global representations linking all visual cues relevant to each response. When such time is not afforded, the system may rely on less efficient strategies such as a hierarchical model. Indeed, the comparison of flat and hierarchical models in this visual task sheds new light to the ancient debate of whether information from distinct sources is integrated before or after the decision stage, analogous respectively to the flat and hierarchical strategies. In multimodal integration such as object localization or motion detection, different modalities provide complementary cues about common objects or features of the environment (object motion, phoneme identity, etc.), so over the lifetime the brain can learn the appropriate crossmodal representations and integrate bimodal information directly over those representations. Indeed, in such context both sources of information are merged prior to decision (*Körding and Wolpert, 2004*). By contrast, when subjects must detect the presence of either a visual or an auditory cue, requires to mapping arbitrarily two distinct unimodal signals into a single response, the relevant crossmodal representations are not formed, and therefore integration could by default only be formed following unimodal decisions, as it has been found experimentally (*Otto and Mamassian, 2012*). One important hypothesis we make is that the level of integration could strongly depend on experience, gradually switching from post-decision (i.e. hierarchical) to pre-decision (i.e. flat) integration.

Finally, an important distinction has to be made between the hierarchical vs. flat nature of the integration process, and the serial vs. parallel nature of the integration process. While the former has to deal with the level and the structure of the representations at which integration takes place before reaching a decision, the latter corresponds to whether evidence from different sources can be integrated at the same time in these representations. The flat model is perfectly compatible with serial integration of evidence, for example if attentional constraints limit the capacity to process evidence from distant visual locations, such as when subjects must integrate from three distinct levels (*Zylberberg et al., 2012*).

In summary, despite premature conclusions, we think that the experimental framework developed by Lorteije and colleagues opens a new promising venue for understanding the level at which evidence is accumulated and individual perceptual decisions are taken in ecological settings.

## Methods

We present here the equations governing the flat race model of decision-making used for simulations. Our implementation of the flat model departs from the one in [*Lorteije et al., 2015*] in the following aspects, which are typically subsumed in many standard implementations: (1) added noise is constant instead of being proportional to signal strength, (2) there is a positive loop (i.e. negative leak) in the integration process and mutual inhibition between competitors and (3) activities cannot be negative (i.e. activation rectification). The variable $x_t^i$ represents the activity of the activation unit $i$ (from 1 to 4) at sample $t$, and evolves according to:

$$\Delta x_t^i = I_0 + \alpha x_{t-1}^i - \beta \sum_{j \neq i} x_{t-1}^j + I_t^i + N(0, \sigma)$$

where $I_0$ is the constant input, $\alpha$ is the auto-excitation term (negative if leaking, positive if positive loop providing bistability), $\beta$ is the inhibition strength, $I_t^i$ is the sensory evidence at sample $t$, and the last term represents white noise of variance $\sigma$. Inhibition is homogeneous and all-to-all between activations units, that is, each unit inhibits equally the other unit from the same branch and the two units

from the alternative branch. Sensory evidence integrates information from the primary and secondary branching points:

$$I_t^i = k_1 a_t^i + k_2 b_t^i$$

$k_1$ and $k_2$ represent the sensitivity to evidence at level 1 and 2 respectively, $a_t^i$ represents the information provided at level 1 in favor of the corresponding branch (i.e. the difference between instantaneous luminosity in that branch and luminosity in the alternative branch at sample $t$), $b_t^i$ represents the information provided at level 2 in favor of the corresponding branch.

We enforce non-negative values for unit activity by using rectification:

$$x_t^i = \max\left(x_{t-1}^i + \Delta x_t^i, \, 0\right)$$

All units are initiated from the same starting point $x_0^i = 0.5$, and the decision is reached whenever any of the units reaches threshold $K$ (**Figure 1C**). If no decision is reached after presentation of all samples, the response corresponding to the unit with highest final activity is selected. Parameters are: input strength at primary branching point k$_1$ = 0.016, at secondary branching points k$_2$ = 0.012, auto-excitation $\alpha$ =0.1, inhibition $\beta$ = 0.07, constant input $I_0$= 0.5, decision threshold = 50, initial values $x_0^i$= 0.5, white noise variance = 1. We chose parameters to reproduce qualitatively the monkey behavioral and neuronal data. We simulated 100,000 trials, which is the same order of magnitude as used in the original experiment. All analyses performed on simulated data strictly reproduced those realized in the original experiment. Results from these analyses are presented in **Figure 2—figure supplement 2**.

In the simulation set of **Figure 2**, designed to reproduce the pattern of reaction times described in the companion paper, we lowered the decision threshold to 30 to produce shorter reaction times. We introduced a minimum integration time of 10 samples (i.e. no boundary for the first 10 samples), consistent with the minimal time of 500 ms after stimulus onset monkeys had to wait before performing a responses saccade. We also changed inhibition to 0.05 and auto-excitation to 0.12. All other parameters were unchanged.

A matlab script for all simulations and analyses is available at http://bit.ly/hyafilmoreno2017.

## Acknowledgements

The authors thank Jan Drugowitsch for his comments on the manuscript.

## Additional information

### Funding

| Funder | Grant reference number | Author |
|---|---|---|
| European Commission | Marie Curie IEF - CEMNET (agreement 629613) | Alexandre Hyafil |
| Ministerio de Economía y Competitividad | PSI2013-44811-P | Rubén Moreno-Bote |

The funders had no role in study design, data collection and interpretation, or the decision to submit the work for publication.

### Author contributions

AH, Conceptualization, Software, Formal analysis, Funding acquisition, Investigation, Visualization, Methodology, Writing—original draft, Project administration, Writing—review and editing; RM-B, Supervision, Investigation, Writing—original draft, Project administration, Writing—review and editing

### Author ORCIDs

Alexandre Hyafil, http://orcid.org/0000-0002-0566-651X

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

## Appendix 1

### Dependence of L2 performance on L1

Lorteije and colleagues defend that a flat model would necessarily yield a dependence of L2 performance on L1 difficulty. In the flat race model described above, the selection signals at branches L2 ($s_t^2 = x_t^1 - x_t^2$) and at L2′ ($s_t^{2'} = x_t^3 - x_t^4$) evolve according to:

$$\Delta s_t^i = (\beta - \alpha)s_t^i + J_t^i + N\left(0, \sqrt{2}\sigma\right),$$

with $J_t^2 = I_t^1 - I_t^2 = 2k_2 b_t^2$ and $J_t^{2'} = I_t^3 - I_t^4 = 2k_2 b_t^{2'}$

The impact of each sample onto L2 selection signal can be directly derived as

$$s_t^i = -2k_2 \sum_{u=1}^{t}(1+\beta-\alpha)^{t-u} b_u^i + N(0, \gamma_t \sigma),$$

where $\gamma_t = \sqrt{2\frac{1-(1+\beta-\alpha)^t}{\alpha-\beta}}$

In other words the selection signals at levels 2 are only influenced by information provided at level two branches $b_t^i$, and not by information provided at level 1 $a_t^i$. It is thus expected as a general result that L2 selection does not depend on L1 difficulty nor any other L1-related variable, contrary to the predictions of Lorteije and colleagues. We detail below four factors that could in principle go against this conclusion. We found however than in practice none induced a significant bias in our simulations:

i.   the above formula is only valid when activation remain above the non-negative boundary. Nevertheless we observed in our simulations that the independence of L2 performance on L1 difficulty still held, even when the zero boundary played an active role (as is the case for the chosen parameter set).

ii.  the decision boundary biases towards earlier choices when L1 task is easy compared to when L1 task is hard. In turn, this would mean lower average integration time for L2 choice and thus a decrement of L2 performance. This is exactly what is observed in the original simulation of the flat race model by Lorteije and colleagues. Evidence for a relatively small decision boundary was taken from the primacy effect, whereby earlier samples had larger impacts onto the decision than later samples in the trial (their **Figure 2E**). However, such primacy effect can alternatively emerge from inhibition, as soon as it outweighs leak: indeed, the weight of each sample $(1+\beta-\alpha)^{t-u}$ decreases when sample position $u$ increases (Suppl. **Figure 1A**). We can thus replicate primacy effect and independence of L2 on L1 difficulty by implementing a relatively high decision threshold, consistent with findings from another fixed-duration perceptual accumulation task (**Brunton et al., 2013**). Indeed, very similar results were observed when we used an infinite threshold.

iii. whether L2 decision corresponds to selection signals at L2 or L2′ depends ultimately on which unit finally wins the race. Thus, while both L2 decision are independent of L1, the *observed* L2 selection signal can be influenced by L1. In practice, we found this had no impact for the parameters reported above and in a large part of the parameter space.

iv.  when additive noise in the units is stimulus-dependent, i.e. if it grows larger when the intensity of the stimulus is larger (our simulations used stimulus-independent noise). In such case, since easier L1 trials correspond to stronger L1 signals, then easier L1 trials also induce larger noise added to the integration process. This could thus deteriorate L2 selection process.

## Appendix 2

### Dependence of selection signals on L1 evidence

When the zero bound for unit activity is hypothesized to play no role, activity for each of the 4 units (TT, TD, DT, DD) at each time step can be decomposed as the sum of a term relative to integration of L1 information, a term relative to integration of L2 information, and a noise term:

$$x_t^{TT} = s_t^1 + s_t^2 + N(0, \gamma_t \sigma)$$

$$x_t^{TD} = s_t^1 - s_t^2 + N(0, \gamma_t \sigma)$$

$$x_t^{DT} = -s_t^1 + s_t^{2'} + N(0, \gamma_t \sigma)$$

$$x_t^{DD} = -s_t^1 - s_t^{2'} + N(0, \gamma_t \sigma)$$

The first term in each equation induces a positive correlation between evidence at L1 and activity at TT and TD units, and a negative one between evidence at L1 and activity at DT and DD units. Furthermore, the design of the experiment with three different difficulty levels for trials creates correlations between evidence at the different levels, as each difficulty level $d$ associated with a given mean $\mu_d$ and standard deviation $\zeta_d$ for the intensity of the individual samples. The correlation across trials between sensory evidence at two different levels (L1 and L2/L2') and at two different sample positions (t and t') is:

$$\rho\left(a_t^i, b_{t'}^i\right) = \frac{Var(\mu_d)}{[Var(\mu_d) + <\zeta_d>]}$$

where $Var$ and $<.>$ represent the variance and mean over the different difficulty levels, respectively. Such correlation between samples increases the correlation between TT activity and L1 evidence, but decreases the correlation between TD and L1 evidence, hence the difference observed in *Figure 3C* (in the other branch, the amplitude of the negative correlation with L1 evidence is enhanced for DD units and decreased for DT units). The other source of heterogeneity of the impact of L1 evidence is the rectification, which impacts more the activity of DD, DT and TD units. Overall, the flat model predicts different levels of modulation of unit activity by evidence in L1 for the four different branches, and an especially low (positive) correlation for the case of the TD unit.

