## [Decision Letter]

Thank you for submitting your article "Breaking down hierarchies in perceptual decision-making" for consideration by *eLife*. Your article has been favorably evaluated by Timothy Behrens (Senior Editor) and three reviewers, one of whom is a member of our Board of Reviewing Editors. The reviewers have opted to remain anonymous.

The reviewers have discussed the reviews with one another and the Reviewing Editor has drafted this decision to help you prepare a revised submission.

Summary:

These two papers present an enlightened and useful discussion about the interpretation of results previously published by Lorteije and colleagues. In that prior study, monkeys performed a task that required them to make a saccadic eye movement to the appropriate endpoint of a visual branching pattern with three bifurcations: one at the top, then two others under each branch, resulting in four distinct endpoints. Each bifurcation had a modulating luminance cue at each branch that determined the correct path: always choose the brighter (on average) branch. Through analyses of behavior and the activity of neurons in cortical areas V1 and V4 whose receptive fields corresponded to the locations of the luminance cues, a primary conclusion from that study was that the monkeys solved the task in a hierarchical manner, with decisions about the top- and lower-level branches first occurring in parallel, then combined for a final choice.

That landmark paper spawned several interesting discussions in the field. Hyafil and Moreno-Bote's technical comment encapsulates one of the critical lines of discussion about whether it is possible to truly distinguish a hierarchical decision-making process from a flat process based on the experimental data of Lorteije and colleagues. It is a critical and complex question. The reply by Zylerberg and colleagues adds further to the discussion, presenting several counter-arguments to the claims of Hyafil and Moreno-Bote.

All three reviewers were impressed by the tone and content of the submissions and agree that both represent worthwhile contributions to the literature. As noted by one of the reviewers, this kind of debate is "very valuable and generally underappreciated."

The above comments are included, verbatim, in our decision letters to both groups of submitting authors. We also will now make both initial submissions available to both groups, so that any potential revisions can fully take into account the claims made in the other submission. We will then allow for one more iteration: if and when we receive revised submissions and deem them appropriate, we will then again make each available to the other group for further revisions and clarifications.

Below are summaries of the discussions among the reviewers that are specific to your submission.

Essential revisions:

The reviewers agreed with the key point, that a "flat" mechanism that includes mutual inhibition can account for many features of the data and generally act like a hierarchical process. However, they also raised several concerns that should be addressed:

1) It would be useful to provide more intuitions about the specific assumptions and parameter values in the flat model that give rise to key types of output. It is critical to note that only some instantiations of the flat model are compatible with the data. The explanation in the legend of Figure 2 is useful but inadequate. Dedicating one or two paragraphs of the main text to model parameters and predictions will make the technical comment much more accessible. For example, why and under what parameter regimes inhibition in a flat model makes the accuracy of L2 decision independent of L1 stimulus strength? Likewise, can you provide better intuitions for the effect of scaling of noise with stimulus strength and rectification of the accumulators? Finally, the flat model explains the late influence of L1 on L2 and L2' choices, removing the need for complex interactions imposed by the hierarchical model. However, it also predicts that different L1 stimulus strengths should cause different magnitudes of suppression on TD neural responses, a prediction that does not match the data, as pointed out by Lorteije et al. It will be useful for readers to know that this prediction stems from Hyafil and Moreno-Bote's assumption that each accumulator equally suppresses the other accumulators. It is likely that suppression is not uniform, causing a TT choice to suppress TD less than DD because TD is a closer option to TT in the decision space than DD. Non-uniform suppression has been reported in various sensory processes and is quite likely to apply to multi-choice decisions.

2) A figure that recapitulates the task and the flat race-mode would also be useful.

3) The authors should be clearer on how they think their model relates to sensory and/or decision circuits. A race-model with mutual inhibition seems plausible in a decision-making area, but in a sensory area? Or do the authors assume the dynamics to play out in higher area and then feed back to V1 and V4? Making their argument explicit in the context of a model like that by Wimmer et al. 2015 would be very helpful. This point also seems strongly related to their claim that Figure 4 in Lorteije et al. argues against the hierarchical model because "localized selection signals merge information provided at different levels." Perhaps these sensory neurons are getting decision-related feedback?

4) The conclusion that the flat model is a "more parsimonious" explanation for the data than the model proposed by Lorteije et al. might be reconsidered, or at least clarified in the context of other lines of evidence from other studies that relate to flat versus hierarchical processing.

5) Zylberberg et al. argue that the RTs produced by the flat model presented by Hyafil and Moreno-Bote were unrealistic (although note that the task did not have a true RT design, and the measured RTs had only a weak dependence on signal strength). Can any version of the flat model (e.g., with collapsing bounds) produce the reported RTs?

6) It would be useful if both model code and relevant data files could be made available.

[Editors' note: further revisions were requested prior to acceptance, as described below.]

Thank you for resubmitting your work entitled "Breaking down hierarchies of decision-making in visual cortex" for further consideration at *eLife*. Your revised article has been favorably evaluated by Timothy Behrens (Senior Editor) and Joshua Gold (Reviewing Editor).

The manuscript has been improved but there are some remaining issues that need to be addressed before acceptance, as outlined below. We also note that our plan is as follows: once we receive an acceptable response to these issues, we will send the updated manuscript to Zylberberg and colleagues, so they have an opportunity to revise their manuscript accordingly. We then will share both papers with both groups, but at that point, if any further changes are desired, they have to be essential and very well justified.

1) Both papers discuss the new analysis by Zylberberg et al. showing that the strength of evidence at L1 affects V4 activity elicited by the L2 and L2' branches. They are in agreement with the finding but disagree with the interpretation. Part of the disagreement appears to stem from misunderstandings from some comments in the Hyafil and Moreno Bote manuscript. Specifically, as quoted by Zylberberg et al., Hyafil and Moreno Bote state in Appendix A that the flat model predicts that "selection signals at levels 2 are only influenced by information provided at level 2 branches […] and not by information provided at level 1." The new analyses by Zylberberg et al. clearly contradict this prediction. However, in their main text, Hyafil and Moreno Bote state that "this observation is most compatible with the flat model and by itself rules out the hierarchical model that relies on complete neural segregation of integration of L1 and L2 evidence." At the very least, please reconcile their statements in the Appendix and the main text.

2) Figure 2 vs. Figure 2—figure supplement 2: why not just show the Figure 2—figure supplement 2 panels in the main figure? Panels H and I could be relegated to the supplement.

---

## [Author Response]

*Essential revisions:*

*The reviewers agreed with the key point, that a "flat" mechanism that includes mutual inhibition can account for many features of the data and generally act like a hierarchical process. However, they also raised several concerns that should be addressed:*

*1) It would be useful to provide more intuitions about the specific assumptions and parameter values in the flat model that give rise to key types of output. It is critical to note that only some instantiations of the flat model are compatible with the data. The explanation in the legend of Figure 2 is useful but inadequate. Dedicating one or two paragraphs of the main text to model parameters and predictions will make the technical comment much more accessible. For example, why and under what parameter regimes inhibition in a flat model makes the accuracy of L2 decision independent of L1 stimulus strength? Likewise, can you provide better intuitions for the effect of scaling of noise with stimulus strength and rectification of the accumulators?*

We agree with the reviewers that the relationship between the observed properties of the modelled network and its underlying ingredients was not clearly presented in the manuscript. We have improved the revised version to point for each observed properties what ingredients are necessary for this to occur: inhibition, rectification, thresholds, etc. Some parts of it were present in supplementary data and has been moved to main text (the rest is in the new Appendix A). Importantly, we have added a table that precisely summarizes this correspondence between features of the model and observed behavioral/neural properties.

*Finally, the flat model explains the late influence of L1 on L2 and L2' choices, removing the need for complex interactions imposed by the hierarchical model. However, it also predicts that different L1 stimulus strengths should cause different magnitudes of suppression on TD neural responses, a prediction that does not match the data, as pointed out by Lorteije et al. It will be useful for readers to know that this prediction stems from Hyafil and Moreno-Bote's assumption that each accumulator equally suppresses the other accumulators. It is likely that suppression is not uniform, causing a TT choice to suppress TD less than DD because TD is a closer option to TT in the decision space than DD. Non-uniform suppression has been reported in various sensory processes and is quite likely to apply to multi-choice decisions.*

We thank the reviewers for the suggestion than non-uniform inhibition in the flat model may allow us to reproduce this observed null effect in the fourth branch (TD). Non-uniform inhibition is indeed a key component of cortical networks that could play a decisive role in this null effect. Actually, we think that the pattern of inhibition should be stronger local inhibition (from TT to TD) may create reduced activity in TD for stronger L1 evidence (because L1 and L2 evidence are correlated, see below), and cancel the positive modulation with L1 evidence observed in the current model.

Even in the current model, we observed that the modulation was not uniform across all branches but was weakest precisely in the TD branch (2.5-4 times weaker than in other branches according to rho’s). As explained in new Appendix B, this effect was caused by the correlation between the strength of L1 and L2 evidence in the design structure of the sensory samples, as a result of the 3 levels of difficulty. As a result, the null statistical effect in TD branch could simply be due to the relative weakness of the modulation in this branch.

This interesting discussion has been added in a new paragraph in main text, complemented by new Appendix B and Figure 3.

Beyond this point, our interpretation of new analysis in Figure 4 of Zylberberg et al. radically diverges from that of its authors. It is indeed the clearest signature of the flat model that each unit integrates information from all available sources prior to decision, and so its activity should be sensitive to both L1 and L2 evidence. The new analysis by Zylberberg and colleagues report effect from single localized populations, analogous to single unit activity from the model, thus perfectly compatible with modulation by L1 information. Thus, we would like to stress that the fact that selection signals in (at least) 3 of the 4 branches are modulated by evidence provided at both L1 and L2 branches is strongly suggestive of the flat model, and directly contradicts the hierarchical model. Indeed, the most essential element of hierarchical model is that evidence from different branches is not integrated within the same neural populations; integration and modulatory effects only occur prior to decision. This discussion has also been added in our manuscript.

*2) A figure that recapitulates the task and the flat race-mode would also be useful.*

Thanks for the suggestion. We have added in Figure 1 representation of the implementation of the flat race-mode as well as one exemplar simulation (Figure 1) In the figure legend, we now point to the original manuscript by Lorteije et al. for further description of the task.

*3) The authors should be clearer on how they think their model relates to sensory and/or decision circuits. A race-model with mutual inhibition seems plausible in a decision-making area, but in a sensory area? Or do the authors assume the dynamics to play out in higher area and then feed back to V1 and V4? Making their argument explicit in the context of a model like that by Wimmer et al. 2015 would be very helpful. This point also seems strongly related to their claim that Figure 4 in Lorteije et al. argues against the hierarchical model because "localized selection signals merge information provided at different levels." Perhaps these sensory neurons are getting decision-related feedback?*

We agree that the point about localized selection signals merging information provided at different levels was not a good one, mostly because it was not clear from that figure whether what is integrated is perceptual evidence or local decisions. The new figure from Zylberberg points to the latter process, as discussed above. The sentence was accordingly removed.

We thank the reviewers for pointing to the Wimmer et al. paper. As we now briefly discuss in the manuscript, it is not clear whether selection signals recorded in visual cortex emerge locally or reflect feedback from higher areas (this is clearly discussed in Lorteije et al. paper). The flat model assumes integration of evidence from across the visual field, and inhibition between the associated representations, and is thus more compatible with the view that integration takes place at a higher cortical area.

*4) The conclusion that the flat model is a "more parsimonious" explanation for the data than the model proposed by Lorteije et al. might be reconsidered, or at least clarified in the context of other lines of evidence from other studies that relate to flat versus hierarchical processing.*

Thanks for the suggestions. We would like to say that we do not mean that the flat model offers a more parsimonious model for all sorts of hierarchical tasks, and thus we have make sure that we do not make such a claim in our manuscript. However, we think that the flat model offers a more parsimonious explanation for the data of the presented experiment, in that all described effects can be explained by classical ingredients of the race model and not ad hoc mechanisms. Following the reviewers’ suggestion, and in light of this comment and the manuscript by Zylberberg et al., we have clarified our point in the final paragraphs of the manuscript about what the comparative advantages of flat vs. hierarchical model, and under what circumstances we hypothesize that each model will prevail.

In particular, we stress now what is believed is an important distinction between the hierarchical vs. flat nature of the integration process, and the serial vs. parallel nature of the timing of integration. The evidence from previous studies related in the first part of Zylberberg et al.’s manuscript only addresses the second question (when integration is serial or parallel), but speaks very little to the underlying structure of the representations that integrate evidence. Thus, we do not rule out that a flat model integrating evidence serially is also present in these other tasks.

*5) Zylberberg et al. argue that the RTs produced by the flat model presented by Hyafil and Moreno-Bote were unrealistic (although note that the task did not have a true RT design, and the measured RTs had only a weak dependence on signal strength). Can any version of the flat model (e.g., with collapsing bounds) produce the reported RTs?*

We had no access to RTs beforehand so did not place much attention to it our first simulations. Actually, as pointed out by the reviewers, this is not a true RT design but rather a mixed RT-fixed duration one, as monkeys had to wait for at least 500 ms (10 samples) before responding, but could wait even more to accumulate more evidence. We have now taken this into account by removing the decision boundary for the first 10 samples, and then a fixed (finite) threshold (Figure 2—figure supplement 2). The value of the finite threshold was adapted to yield approximately the same mean reaction times depending on task difficulty as reported by Zylberberg et al. Apart from the reported RTs, all previously reported behavioral and neural effects were reproduced, including the lack of modulation of L2 performance by L1 difficulty (Figure 2—figure supplement 2).

We would like to add that Zylberberg et al. report that after a simple modulation of a fixed threshold in the flat model (unlike the double threshold strategy we implemented), L2 performance was not independent anymore or L1 difficulty. We believe that the discrepancies between their simulations and ours stems from the distribution of early responses in the two models. As commented in point 1, modulation of L2 performance by L1 difficulty emerges mostly when L1 difficulty has a large impact on reaction times, and hence on the timing of L2 integration. With a single fixed threshold as used in simulation by Zylberberg and colleagues, there is a significant portion of easy L1 trials with early responses (RT<500 ms), which robustly degrades L2 performance. By contrast, by ensuring that decisions integrate at least the 10 first samples, we make sure that there is a minimum integration window for L2 decision. Note that this time-dependent bound is a form of collapsing bound, so we agree with the reviewer that indeed that is a plausible mechanism that can explain the observed RTs. We would like to conclude by saying that we have provided the results for the modified version of the network as supplementary figure, and left the original as main figure. We think this presentation better reflects the discussion that has taken place between the two manuscripts. We could also provide the new results in the main figures in place of the former ones, if judged more adequate by editors and reviewers.

*6) It would be useful if both model code and relevant data files could be made available.*

This is a good point. We are working on adapting the code to be readily usable by any matlab user. It will comprise both generation of synthetic data and its subsequent analysis (there is no middle way data files). It will be accessible within the next few days at the following link: bit.ly/flatdecision

[Editors' note: further revisions were requested prior to acceptance, as described below.]

*[…] 1) Both papers discuss the new analysis by Zylberberg et al. showing that the strength of evidence at L1 affects V4 activity elicited by the L2 and L2' branches. They are in agreement with the finding but disagree with the interpretation. Part of the disagreement appears to stem from misunderstandings from some comments in the Hyafil and Moreno Bote manuscript. Specifically, as quoted by Zylberberg et al., Hyafil and Moreno Bote state in Appendix A that the flat model predicts that "selection signals at levels 2 are only influenced by information provided at level 2 branches […] and not by information provided at level 1." The new analyses by Zylberberg et al. clearly contradict this prediction. However, in their main text, Hyafil and Moreno Bote state that "this observation is most compatible with the flat model and by itself rules out the hierarchical model that relies on complete neural segregation of integration of L1 and L2 evidence." At the very least, please reconcile their statements in the Appendix and the main text.*

We would like to thank the reviewers for making this point, as indeed there has been some confusion between our two manuscripts.

Specifically, we believe that Zylberberg et al. made a confusion between two things: “unit activity”, i.e. the localized selection signal in each of the four units; and “selection signals for L2” that we compute by taking the difference between the activities of two units within the same branch. While in the latter the influence of L1 evidence is effectively removed (as described in Annex A), in the former influence of L1 is essential: it is indeed the clearest signature of the flat model that each unit integrates information from all available sources prior to decision, and so its activity should be sensitive to both L1 and L2 evidence. The new analysis by Zylberberg and colleagues report effect from single localized populations, analogous to single unit activity from the model, in which we indeed predict modulation by L1 information.

We have now made this important distinction clearer in the manuscript, by adding the following note in the corresponding section:

“At this point, an important distinction has to be made in the hierarchical model between localized activations, which indeed mix evidence from both branches, and selection signals at L2, extracted by looking at the difference between two units in the same L1 branch, and which as shown above are largely insensitive to L1 signals.”

*2) Figure 2 vs. Figure 2—figure supplement 2: why not just show the Figure 2—figure supplement 2 panels in the main figure? Panels H and I could be relegated to the supplement.*

Actually we could not decide in our previous submission which was the better presentation: to put either the original analysis (with incorrect reaction times distribution) or the new analysis in the main figure. We thought that the former solution would allow to present the very data that Zylberberg et al. are commenting on in the companion paper. But we agree that the alternative option that the reviewers present is more straightforward. Changes were made accordingly.